# Elucidating Circular Ribonucleic Acid Mechanisms Associated with Splicing Factor 3 Inhibition in Cervical Cancer

**DOI:** 10.3390/ijms262210883

**Published:** 2025-11-10

**Authors:** Amahle Nyalambisa, Babatunde Adebola Alabi, Zodwa Dlamini, Rahaba Marima

**Affiliations:** 1Pan African Cancer Research Institute (PACRI), DSI/NRF SARChI Chair in Precision Oncology and Cancer Prevention, SAMRC Precision Oncology Research Unit (PORU), University of Pretoria, Hatfield, Pretoria 0028, South Africa; u23013730@tuks.co.za (A.N.); babatunde.alabi@up.ac.za (B.A.A.); zodwa.dlamini@up.ac.za (Z.D.); 2Department of Medical Oncology, Steve Biko Academic Hospital, University of Pretoria, Hatfield, Pretoria 0028, South Africa

**Keywords:** cervical cancer, circular ribonucleic acids, splicing factor 3, *hsa_circ_0001038*, *circRNA_400029*, alternative splicing

## Abstract

Cervical cancer (CCa) is the fourth leading cause of cancer-related deaths among women worldwide, with nearly 90% of cases in low- and middle-income countries, especially in Sub-Saharan Africa. This study explores the roles of circular ribonucleic acids (circRNAs), *hsa_circ_0001038 and circRNA_400029*, and the impact of the serine/arginine-rich splicing factor 3 (SRSF3) inhibitor, theophylline, in CCa cell lines. We utilized cell cycle fluorescence-activated cell sorting (FACS) and Annexin V/propidium iodide (PI) assays to evaluate theophylline’s effects on SiHa and C33A cell lines. Results showed S-phase arrest in SiHa and G2/M arrest in C33A, with significant cytotoxic effects indicated by apoptosis analysis. Using CircAtlas, we identified micro ribonucleic acids (miRNAs) binding to *hsa_circ_0001038*, particularly *miR-205-5p*, which has a tumour-suppressive role. miRTarBase identified *miR-16-5p* as a key interacting miRNA for *circRNA_400029*. We constructed a competing endogenous ribonucleic acid (ceRNA) network, revealing multiple miRNA targets. Pathway analysis via the Kyoto Encyclopedia of Genes and Genomes (KEGG) highlighted critical signalling pathways involved in CCa oncogenesis. In conclusion, theophylline demonstrates cytotoxicity in CCa cells, suggesting its potential for repurposing in CCa theranostics, though further optimization is necessary.

## 1. Introduction

Cervical cancer (CCa) is the fourth most prevalent cause of cancer-related deaths among women worldwide [1]. Nearly ninety percent of CCa cases occur in low- and middle-income countries (LMICs), with Sub-Saharan Africa (SSA) accounting for over eighty percent of the global burden [2]. Numerous environmental and genetic risk factors are associated with CCa, including smoking, alcohol use, multiple sexual partners, unprotected intercourse, low socioeconomic status, a family history of CCa, prolonged oral contraceptive use, human immunodeficiency virus (HIV) infection, and persistent human papillomavirus (HPV) infection [3]. Persistent HPV infection, in particular, is the primary risk factor for CCa, with HPV 16 responsible for approximately 55–60% of cases, followed closely by HPV 18 [4]. Conventional treatments for CCa include surgery, radiation therapy, chemoradiation, external beam radiation therapy (EBRT), chemotherapy, and brachytherapy. However, these methods often fail to achieve the desired outcomes and can result in significant adverse effects [5,6,7]. Therefore, identifying novel diagnostic biomarkers and promising therapeutic targets is crucial for the effective detection and eradication of tumour cells in cervical tissues.

Noncoding ribonucleic acids (ncRNAs) have been shown to play key roles in cancer progression and may serve as potential therapeutic targets to improve CCa prognosis [8,9,10]. In mammalian cells, noncoding transcripts constitute approximately 99% of total ribonucleic acid (RNA) [11,12]. Circular ribonucleic acids (circRNAs), a subtype of ncRNAs, are characterized by their covalently closed structure, which renders them resistant to exonuclease degradation [13,14]. CircRNAs are typically located in the cytoplasm of eukaryotic cells [15]. Unlike canonical splicing of linear RNA, circRNAs are produced through back-splicing, which covalently joins a downstream 3′ splice donor site to an upstream 5′ splice acceptor site, forming a closed loop [16,17]. The serine/arginine-rich splicing factor 3 (SRSF3) has been shown to play a role in this process in other cancers [18].

SRSF3 is a member of the serine/arginine-rich (SR) protein family—an important group of proteins that play a pivotal role in CCa development by influencing RNA splicing [19]. Within this family, SRSF3, the smallest member, is highly expressed in CCa and significantly contributes to tumourigenesis. However, the regulatory mechanisms involving SRSF3 and circRNAs in CCa remain poorly understood. CircRNAs have been implicated in various cancers as either oncogenic or tumour-suppressive molecules, making them potential targets for novel therapeutic strategies. For instance, *circRNA_400029* is upregulated in cervical cancer and promotes cell proliferation, migration, and invasion while inhibiting apoptosis via the *miR-1285-3p*/TLN1 axis [20]. Similarly, hsa_circ_0001038 is overexpressed in cervical cancer and promotes proliferation, migration, and invasion by acting as a ceRNA for *miR-337-3p* [21]. Moreover, circRNA_101996 has been reported to accelerate cervical cancer development through regulation of the *miR-1236-3p*/TRIM37 axis, and circ_0084927 facilitates tumour progression by sponging *miR-142-3p* and upregulating ARL2 [22]. Conversely, circ_0067934 silencing induces apoptosis and cell-cycle arrest, while circ_0000515 downregulation suppresses proliferation through the *miR-326*/SOX11 axis [23].

Theophylline, an SRSF3 inhibitor, has been identified as a potential therapeutic agent for investigating the effects of SRSF3 suppression in CCa cells [24]. While SRSF3 has been associated with the regulation of key oncogenic pathways in CCa, its interaction with circRNAs remains largely unexplored. SRSF3 inhibition could potentially alter circRNA expression patterns, leading to changes in cell-cycle regulation, apoptosis, and cancer cell proliferation. By evaluating circRNA expression profiles and their roles in cellular pathways following SRSF3 inhibition, this study aims to provide new insights into the therapeutic potential of targeting splicing factors and circular RNAs in CCa. Understanding these molecular interactions could enhance treatment strategies and potentially support the repurposing of theophylline for CCa therapy.

## 2. Results

### 2.1. Theophylline Alters Cell Cycle Progression in CCa Cells

Theophylline treatment induces significant alterations in cell cycle distribution in both SiHa and C33A CCa cell lines (Figure 1A). In C33A, at 24 h, the G0/G1 population significantly decreased from 72.55% (Figure 1A) to 50.68% (Figure 1B) (*p* = 0.0053), while the G2/M population significantly increased from 9.03% (Figure 1A) to 34.88% (Figure 1B) (*p* = 0.0018), suggesting a G2/M phase arrest. At 48 h, the G0/G1 population modestly increased from 42.23% (Figure 1E) to 49.45% (Figure 1F), and the S-phase decreased from 20.15% (Figure 1E) to 14.32% (Figure 1F); however, these changes were not statistically significant (*p* = 0.147 for G0/G1; *p* = 0.231 for S). In the SiHa cell line at 24 h, a significant reduction in the G0/G1 phase was observed, from 75.20% (Figure 1C) to 29.95% (Figure 1D) (*p* < 0.0001), alongside a significant increase in the S-phase population from 15.49% (Figure 1C) to 66.40% (Figure 1C) (*p* < 0.0001), indicating strong S-phase arrest. By 48 h, this effect intensified, with G0/G1 phase dropping from 71.13% (Figure 1G) to 8.20% (Figure 1H) (*p* < 0.0001) and S-phase cells increasing from 22.05% (Figure 1G) to 87.51% (Figure 1H) (*p* < 0.0001).

### 2.2. Theophylline Induces Apoptosis in CCa Cell Line SiHa and C33A

Theophylline treatment significantly induced apoptosis in both SiHa (Figure 2, Panel A) and C33A (Figure 3, Panel A) cervical cancer cell lines. In both models, there was a marked increase in apoptotic populations compared to the control treatments. At 24 h, theophylline exhibited a notable apoptotic effect that was comparable to cisplatin (12 μM) in C33A cells but slightly less pronounced in SiHa cells. By 48 h, the apoptotic response to theophylline was markedly enhanced in both cell lines, demonstrating a clear time-dependent effect. In SiHa, theophylline treatment significantly increased late apoptotic populations compared to controls at both time points. Although the apoptotic response in SiHa cells was slightly lower than that induced by cisplatin, it remained statistically significant (*p* < 0.0001 vs. vehicle control; Figure 2, Panel B) and consistent over time. In C33A, theophylline exerted a significant increase in late apoptotic cells at both 24 and 48 h, exceeding levels observed in the vehicle control and surpassing even those induced by cisplatin (*p* < 0.0001 vs. vehicle control; *p* < 0.01 vs. cisplatin; Figure 3, Panel B).

Flow cytometric profiles further revealed a distinct rise in Annexin V^+^/PI^−^ cells at 24 h, indicating that theophylline initially triggers early apoptosis before progressing to the late apoptotic phase at 48 h. The increase in Annexin V^+^/PI^+^ populations at later times likely reflects secondary necrosis accompanying advanced apoptotic stages, rather than primary necrosis. These findings collectively suggest that theophylline induces a genuine apoptotic cascade that evolves, leading to sustained cytotoxic effects in both CCa cell lines.

### 2.3. In Silico Bioinformatics Analysis of Differentially Expressed circRNAs hsa_circ_0001038 & circRNA_400029 in CCa

#### 2.3.1. Prediction of circRNA_400029 miRNA Interactions Using miRTarBase

MiRTarBase, a database of validated miRNA-target interactions, identified potential miRNAs targeting circRNA_400029. The analysis revealed that circRNA_400029 originates from the *ribosomal protein L13* (*RPL13*) gene and interacts with four specific miRNAs (Figure 4). Among these, miR-16-5p was selected for further investigation. As a member of the miR-16 family, miR-16-5p plays a role in multiple molecular pathways linked to CCa progression. Notably, it functions as a tumour suppressor in CCa, with the potential to inhibit tumour growth and angiogenesis.

#### 2.3.2. ceRNAs Network of circRNA_400029—miRNA Using miRTarBase

The miRTarBase was used to identify ceRNAs for circRNA_400029. The analysis identified several hundred predicted downstream gene targets of miR-16-5p. It was found that a single gene can be targeted by multiple miRNAs within the network, and vice versa (Figure 5). For instance, cyclin-dependent kinase 6 (CDK6), *G1/S-specific cyclin-D1* (*CCND1*), *High Mobility Group AT-Hook 1* (*HMGA1*), and *High Mobility Group AT-Hook 2* (*HMGA2*) interact with both hsa-miR-16-5p and hsa-miR-26a-5p. On the other hand, miR-744-5p and miR-615-3p did not share any common mRNA targets with other miRNAs in the ceRNA network. The mRNA targets available in the database depend on experimentally validated data uploaded by the research community, so the absence of shared targets for miR-744-5p and miR-615-3p may be due to limited research on these miRNAs rather than a lack of actual interactions.

#### 2.3.3. MiR-16-5p—Gene Target Pathway Analysis Using KEGG Pathway

The miR-16-5p gene target pathway analysis showed that miR-16-5p targets *WNT4*, a key ligand in the *Wnt/β-catenin* signalling pathway. KEGG pathway analysis indicated that downstream components of the *Wnt/β-catenin* pathway, including *FZD*, *LRP5/6*, *DVL*, and *FRAT,* are potentially upregulated in association with increased *WNT4* expression. Additionally, *SRSF3* was identified as a downstream target of *β-catenin*/*TCF4* transcriptional activity. This suggests that circRNA_400029 may promote CCa progression by targeting the miR-16-5p/*Wnt/β-catenin*/*SRSF3* axis.

#### 2.3.4. Prediction of hsa_circ_0001038—miRNA Interactions Using circAtlas

CircAtlas predicted a total of 23 miRNAs (Figure 6) to have a binding spot for hsa_circ_0001038. MiR-205-5p was then chosen for further analysis. MiR-205-5p is a tumour-suppressive miRNA that belongs to the miR-200 family, which is known for its role in regulating epithelial to mesenchymal transition (EMT) and influencing CCa progression [25,26].

#### 2.3.5. ceRNA Network of *miR-205-5p* Using miRTarBase

MiRTarBase was then used to explore the potential downstream targets of *miR-205-5p*. The database returned several hundred predicted gene targets. To narrow this down, only the gene targets linked to CCa pathogenesis were selected for inclusion in the ceRNA network (Figure 7).

Among the identified gene targets, *DEAD-box helicase 5* (*DDX5*) emerged as a key gene of interest due to its known involvement in cancer-related pathways. *DDX5* was predicted to be a downstream effector regulated by the circRNA hsa_circ_0001038 via the miR-205-5p/SRSF3/*DDX5* axis. Interaction analyses suggested that hsa_circ_0001038 may act as a molecular sponge for miR-205-5p, a tumour-suppressive miRNA, thereby potentially modulating *DDX5* expression. Additionally, *SRSF3* was identified as a regulatory splicing factor influencing *DDX5* isoform expression. Notably, samples with elevated *SRSF3* expression also exhibited increased levels of the *DDX5-L* isoform, consistent with previous findings that link this variant to oncogenic signalling. These findings support a putative regulatory relationship involving hsa_circ_0001038, miR-205-5p, *SRSF3*, and *DDX5* in CCa.

## 3. Discussion

CCa remains one of the most prevalent causes of cancer-related mortality in women, particularly in LMICs, where it accounts for a disproportionate global burden [2]. Sub-Saharan Africa alone contributes over 80% of global CCa cases [3], underscoring the need for innovative diagnostic and therapeutic approaches that are both effective and accessible. In recent years, circRNAs have emerged as important regulatory molecules in various cancers, including CCa [27], particularly hsa_circ_0001038 [28] and circRNA_400029 [29]. CircRNAs are generated through back-splicing mechanisms, with *SRSF3* playing a pivotal role in their biogenesis in other cancers [23]. However, the functional implications and downstream regulatory networks involving these circRNAs in CCa remain poorly understood. This study investigated the effects of splicing factor 3 inhibition using theophylline, a methylxanthine derivative known to modulate pre-mRNA splicing, on two cervical cancer (CCa) cell lines—HPV-16-positive SiHa and HPV-negative C33A. Cell cycle and apoptosis analyses revealed that theophylline exerts both cytostatic and cytotoxic effects in a cell line–dependent manner. Specifically, theophylline induced S-phase arrest in SiHa and G2/M arrest in C33A, indicating disruption of key cell cycle checkpoints and replication control. These alterations were accompanied by a significant increase in apoptotic cell populations, with effects comparable to or exceeding those of cisplatin, a standard chemotherapeutic agent. Importantly, although a marked elevation in late apoptotic (Annexin V^+^/PI^+^) populations was observed, this may partly reflect secondary necrotic events that occur following the progression of apoptosis, especially given the relatively high concentration (10 mM) of theophylline used. Therefore, the primary theophylline-induced effect is more accurately attributed to the activation of early apoptotic signalling rather than direct necrosis. The significant rise in Annexin V^+^/PI^−^ cells at 24 h further supports this interpretation, indicating that apoptosis is initially triggered before membrane disruption occurs. Previous studies have demonstrated that inhibition of splicing factors can lead to cell cycle arrest and apoptosis across various tumour models. For example, Zhang et al. [30] reported that inhibition of SF3B1 induced G2/M phase arrest and apoptotic signalling in gastric cancer cells, consistent with our findings in C33A. Similarly, Pérez-Pérez et al. [31] showed that methylxanthines such as theophylline disrupt cell cycle progression and trigger apoptosis in glioblastoma cells, supporting our observation of S-phase arrest in SiHa. Collectively, our findings highlight that theophylline induces early apoptotic responses and cell cycle dysregulation through potential interference with splicing-dependent regulators, underscoring its promise as a repurposed therapeutic candidate in CCa. Future dose–response studies using lower concentrations will be necessary to delineate the threshold between apoptotic and necrotic outcomes and to optimize its therapeutic window.

Moreover, the relationship between cytotoxicity and RNA integrity has been explored in the context of splicing factor inhibition. Previous reports by Petasny et al. [32] and Bradley et al. [33] have revealed that splicing factor disruption not only affects cell cycle progression but also impacts RNA stability, complicating downstream analyses. Our findings align with these observations, suggesting that the cytotoxic effects of theophylline may extend beyond immediate cellular responses to include detrimental impacts on RNA quality. In light of these challenges, future studies should consider alternative methodologies, such as single-cell RNA sequencing or RNA-seq, which may provide insights into gene expression changes without the constraints imposed by RNA integrity issues. Balancing therapeutic potency with cell viability to maintain molecular integrity is a common challenge in translational cancer research. However, the observed cell death underscores the therapeutic potential of theophylline in cancer.

To further explore the molecular mechanisms potentially regulated by theophylline, bioinformatics analyses revealed that MiR-16-5p targets several oncogenic mRNAs such as *CDK6*, *CCND1*, and *HMGA2*, which are implicated in unchecked cell proliferation and chromatin remodelling. Notably, many of these targets are components of the *Wnt/β-catenin* signalling pathway, a pathway commonly activated by HPV oncogenes during cervical tumorigenesis. By sponging miR-16-5p, circRNA_400029 may contribute to oncogenic signalling cascades, promoting tumour survival and progression. Similarly, hsa_circ_0001038 was predicted to sponge miR-205-5p, another tumour-suppressive miRNA involved in the inhibition of epithelial-to-mesenchymal transition (EMT) and metastasis. The downregulation of miR-205-5p has been linked to increased expression of *DDX5* [34], a cofactor of *SRSF3* that promotes oncogenic alternative splicing and activates the *PI3K*/*AKT* signalling pathway. This regulatory axis supports CCa cell proliferation and resistance to apoptosis. The mutual reinforcement between *SRSF3*, *DDX5*, and their downstream targets suggests a feedforward loop, potentially exacerbated by circRNA-mediated miRNA sequestration. These ceRNA networks may be central to cervical tumorigenesis and could represent novel targets for therapeutic intervention. Additionally, pathway enrichment analysis using KEGG highlighted several signalling axes associated with CCa, including the HPV *E5*–*EGFR*–*RAS*–*ERK* and HPV *E6*–*Notch* pathways, which are known to promote oncogenesis through enhanced cell proliferation, survival, and splicing deregulation. The overlap between these signalling pathways and those regulated by miR-16-5p and miR-205-5p further supports the functional relevance of circRNA_400029 and hsa_circ_0001038 in CCa biology.

## 4. Materials and Methods

### 4.1. Cell Culture and Drug Treatment

CCa cell lines SiHa (HPV 16+) and C33A (HPV^−^) were purchased from the American Type Culture Collection (ATCC, Manassas, VA, USA). SRSF3 inhibitor (Theophylline) was purchased from MedChemExpress (Monmouth Junction, NJ, USA). Cisplatin was purchased from Sigma-Aldrich^®^ Solutions (St. Louis, MO, USA). The human CCa cell lines SiHa (HPV 16+) and C33A (HPV^−^) were cultured in Dulbecco’s Modified Eagle’s Medium (DMEM: F12) (Thermo Fisher Scientific, Waltham, MA, USA). This medium was enriched with 10% (*v*/*v*) fetal bovine serum (FBS) from Gibco (Thermo Fisher Scientific, Waltham, MA, USA) and supplemented with 1% streptomycin (Thermo Fisher Scientific, Waltham, MA, USA). The cells were maintained at 37 °C in a humidified incubator with 5% CO_2_ until they reached 70% confluency. Before theophylline treatment, the cells were subjected to serum starvation for 24 h to synchronize the cell cycle.

### 4.2. Propidium Iodide (PI) Cell Cycle FACS Analysis

The C33A and SiHa cells were treated with 10 mM theophylline for 24 and 48 h, with 0.1% DMSO as a control. Cells were cultured at a density of 5 × 10^5^ cells per well in 6-well plates and harvested by centrifugation at 500× *g* for 5 min. The supernatant was discarded, and cells were washed twice with ice-cold 1× PBS. The cells were fixed by adding 7 mL of ice-cold 70% ethanol while vortexing and stored at −20 °C for 24 h. After fixation, cells were centrifuged again, washed with PBS, and resuspended. Cells were stained with a solution containing propidium iodide (PI) and RNase A, achieving final concentrations of 1 µg/mL of PI and 0.2 mg/mL of RNase A. The staining solution was incubated for 30 min at 37 °C, then placed on ice. The samples were analyzed by flow cytometry (Beckman Coulter CytoFLEX cytometer) (Beckman Coulter, Brea, CA, USA) using Kaluza analysis software version 2.1 (Beckman Coulter, Brea, CA, USA)within an hour, and the results were presented as histograms.

### 4.3. Apoptosis Assay by FACS

C33A and SiHa cells were cultured separately and treated with 10 mM theophylline for 24 and 48 h [24]. Cisplatin (12 μM) served as a positive control, while DMSO was used as a negative control. After treatment, cells were digested with EDTA-free trypsin and centrifuged at 300× *g* for 5 min at 4 °C. The cells were then incubated under optimal culture conditions for 30 min. Cells were washed twice with ice-cold PBS, centrifuged again at 300× *g* for 5 min at 4 °C, and resuspended in 100 μL of 1× binding buffer at a density of 1–5 × 10^5^ cells. To each tube, 4–5 μL of CL488-Annexin V and 5 μL of PI working solution were added. Compensation adjustments were made using additional tubes with single dyes (either CL488-Annexin V or PI). Samples were incubated for 10–15 min at room temperature in the dark. After incubation, 400 μL of 1× binding buffer (or 1× PBS) was added to each tube. The samples were analyzed using the Beckman Coulter CytoFlex flow cytometer within an hour, and the results were presented as histograms.

### 4.4. Bioinformatics Analysis

To determine the interactions of the hsa_circ_0001038 and circRNA_400029, bioinformatics tools including CircAtlas, miRTarBase, and KEGG pathway were used. CircAtlas was used to predict miRNA binding sites of hsa_circ_0001038. However, circRNA_400029 could not be found in CircAtlas. This could be because this database relies on experimental data uploaded by the research community, and this circRNA may not have been included in the available datasets. Therefore, to explore miRNA interactions for circRNA_400029, MiRTarBase was used. MiRTarBase was also used to visualize the ceRNA network of the two circRNAs associated with miRNAs and mRNAs, which were identified using circAtlas for hsa_circ_0001038 and miRTarBase for circRNA_400029. KEGG pathway was used to map the miRNA-gene target pathway.

### 4.5. Statistical Analysis

Data acquired from at least 3 independent experiments, each in at least triplicate, was analyzed using GraphPad Prism version 9.0 statistical software. Statistical significance was performed using ONE-WAY ANOVA followed by Tukey’s post hoc test for multiple comparisons. The significance level was deemed at *p* < 0.05. The results are presented as mean ± standard error of the mean (SEM).

## 5. Conclusions and Future Perspective

This study demonstrated that *SRSF3* inhibition using theophylline exerts potent cytostatic and pro-apoptotic effects in CCa cells by interfering with cell cycle progression and inducing apoptosis. Bioinformatics analysis further implicates circRNA_400029 and hsa_circ_0001038 as key regulators in CCa through their interaction with tumour-suppressive miRNAs and downstream ceRNA oncogenic pathways. Future research should include dose–response studies with lower theophylline concentrations to confirm its selective apoptotic effects and minimize necrosis. Functional validation of circRNA_400029–miR-16-5p and hsa_circ_0001038–miR-205-5p interactions through knockdown or overexpression experiments will help clarify their regulatory roles. Integrating RNA sequencing and proteomic analyses could further elucidate how theophylline modulates SRSF3-dependent pathways and downstream signalling networks. Finally, in vivo and organoid-based studies are essential to evaluate the therapeutic relevance of theophylline and the biomarker potential of these circRNAs in cervical cancer diagnosis and treatment response.

## Figures and Tables

**Figure 1 ijms-26-10883-f001:**
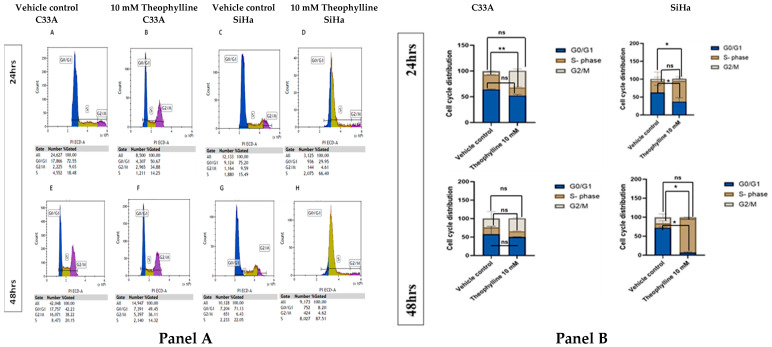
(**Panel A**) Flow cytometry analysis of cell cycle distribution in CCa cell lines C33A and SiHa following treatment with vehicle control or 10 mM theophylline for 24 and 48 h. Subpanels (**A**–**D**) depict C33A cells at 24 h (**A**,**B**) and SiHa cells at 24 h (**C**,**D**), while subpanels (**E**–**H**) show C33A cells at 48 h (**E**,**F**) and SiHa cells at 48 h (**G**,**H**). (**Panel B**) Quantification of cell cycle phase distribution (G0/G1, S, and G2/M) presented as stacked bar graphs for both C33A and SiHa cells under each treatment condition. Data are expressed as mean ± SEM from three independent experiments. Statistical significance was determined using one-way ANOVA followed by Tukey’s post hoc test for multiple comparisons. Statistical differences are denoted as: (*) *p* < 0.05, (**) *p* < 0.01; non-significant differences are indicated as “ns”.

**Figure 2 ijms-26-10883-f002:**
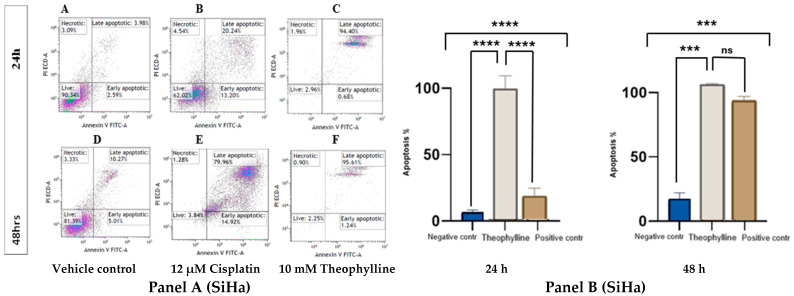
(**Panel A**) Effects of theophylline (10 mM) on CCa cell line SiHa at 24 (**A**–**C**) and 48 h (**D**–**F**). Cells were treated with either vehicle control (**A**,**D**), positive control (12 μM of cisplatin) (**B**,**E**) or 10 mM of theophylline (**C**,**F**) for 24 and 48 h. (**Panel B**) Bar graphs showing the apoptotic effects of 10 mM theophylline on SiHa cells at 24 and 48 h. Statistical significance was performed using ONE-WAY ANOVA followed by Tukey’s post hoc test for multiple comparisons. Statistical analysis is indicated by asterisks, (***) *p* < 0.001, (****) *p* < 0.0001; non-significant differences are indicated as “ns”. Data is represented as ±SEM.

**Figure 3 ijms-26-10883-f003:**
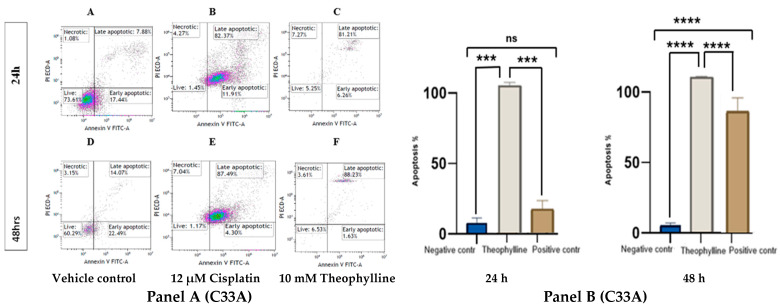
(**Panel A**) Effects of theophylline (10 mM) on CCa cell line C33A at 24 (**A**–**C**) and 48 h (**D**–**F**). Cells were treated with either vehicle control (**A**,**D**), positive control (12 μM of cisplatin) (**B**,**E**), or 10 mM of theophylline (**C**,**F**) for 24 and 48 h. (**Panel B**) Bar graphs showing antiproliferative effects of 10 mM theophylline on C33A cells at 24 and 48 h. Statistical significance was performed using ONE-WAY ANOVA, followed by Tukey’s post hoc test for multiple comparisons. Statistical analysis is indicated by asterisks, (***) *p* < 0.001, (****) *p* < 0.0001; non-significant differences are indicated as “ns”. Data is represented as ±SEM.

**Figure 4 ijms-26-10883-f004:**
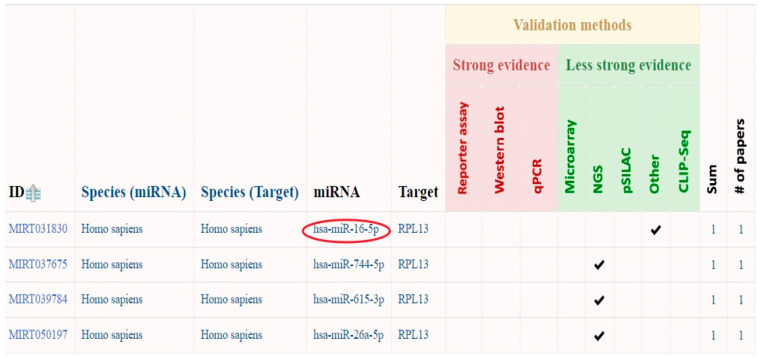
CircRNA_400029 interacting miRNAs. Colored symbols represent the type and validation status of circRNA–miRNA interactions. Red circles indicate experimentally validated miRNA interactions retrieved from miRTarBase, while check marks denote predicted interactions supported by computational evidence. Color shading reflects interaction strength or confidence scores, with darker tones indicating higher confidence levels.

**Figure 5 ijms-26-10883-f005:**
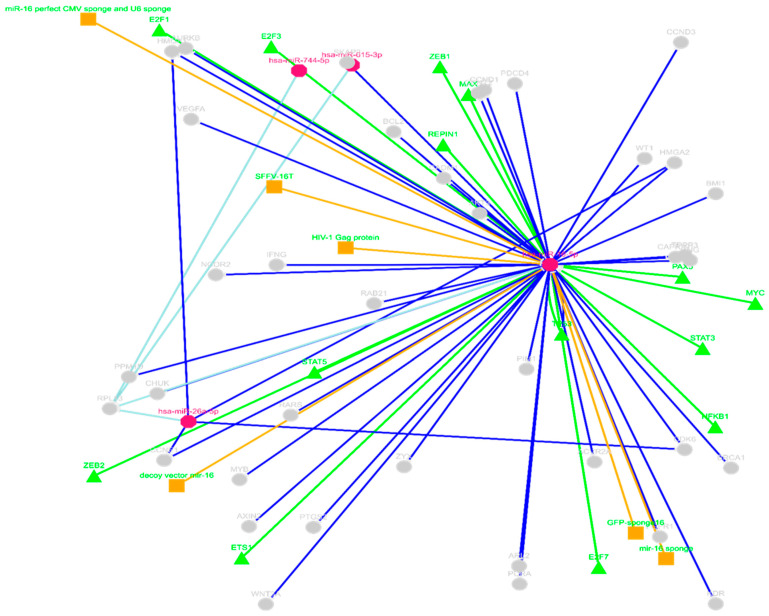
The competing endogenous RNA (ceRNA) network of miR-16-5p using miRTarBase. The miRNAs are represented in a pink Octagon, and the gene targets are represented in grey circles, while the green triangles represent regulatory proteins, and the orange squares represent viral proteins.

**Figure 6 ijms-26-10883-f006:**
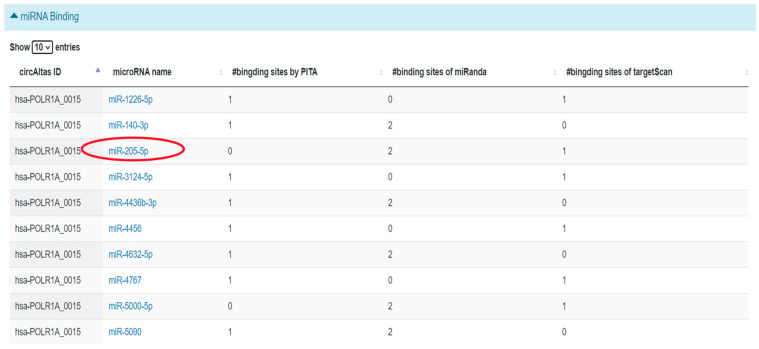
Predicted miRNA binding of *hsa_circ_0001038* using circAtlas. Red circles indicate predicted miRNA binding sites for *hsa_circ_0001038* as identified by CircAtlas. Each red circle corresponds to a miRNA predicted to interact with *hsa_circ_0001038* based on sequence complementarity and binding energy scores.

**Figure 7 ijms-26-10883-f007:**
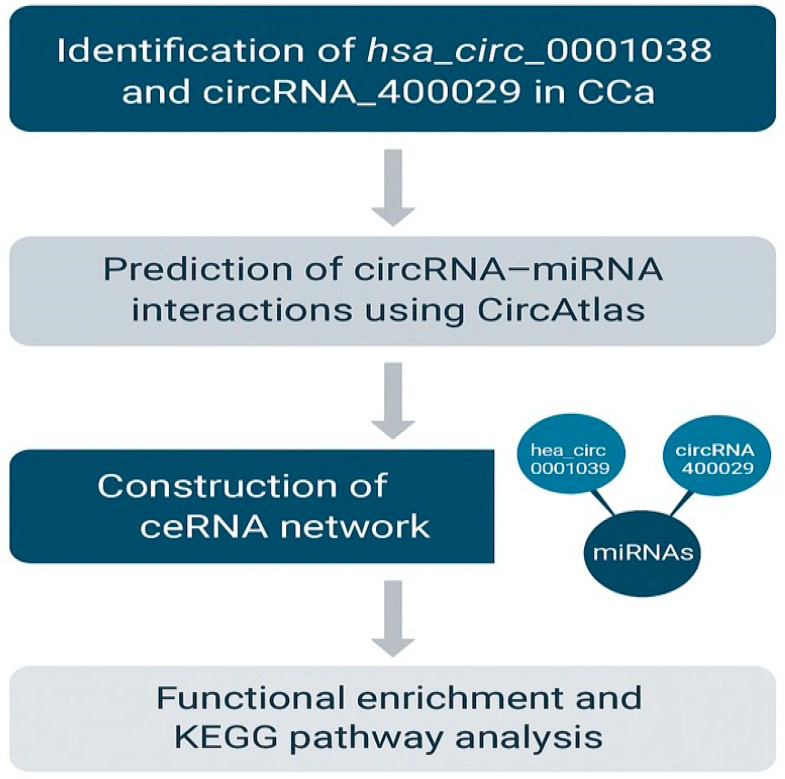
Summarized illustration of the bioinformatics findings of this study. CircRNA_400029 and hsa_circ_0001038 are highly expressed in CCa cells, associated with CCa-related miRNAs, respectively. These circRNAs form ceRNA networks with multiple CCa-related miRNAs and mRNAs, sponging tumour-suppressive miRNAs to promote CCa progression.

## Data Availability

The data supporting this study’s findings are available from the corresponding author, [RM], upon reasonable request.

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
