# Peer review of "Elucidating Circular Ribonucleic Acid Mechanisms Associated with Splicing Factor 3 Inhibition in Cervical Cancer"

_ijms, 2025, doi:10.3390/ijms262210883_

Round 1

Reviewer 1 Report

Comments and Suggestions for Authors

This manuscript appears to present only preliminary findings. It does not provide significant new insights or in-depth mechanistic studies regarding the antitumor or apoptotic effects of Theophylline in vitro. The experimental design is inappropriate and lacks essential data to support the conclusions in this manuscript.

  1. The authors should clarify whether Theophylline affects cell viability and cytotoxicity in both human cervical cancer cells and normal cervical cells. It is recommended to determine and present the ICâ‚…â‚€ values for both cell types to assess the drug’s selectivity and potency.
  2. Please verify whether the concentration of Theophylline (10 mM) is correct. This appears to be a relatively high dose that may induce cytotoxicity rather than anticancer effects. Such a concentration may not be appropriate for evaluating the potential of Theophylline as an anticancer agent in human cervical cancer.
  3. The overall manuscript formatting does not follow scientific writing conventions. For instance, sections such as Figure 1, Figure 2, and their corresponding legends are missing or incorrectly formatted.
  4. The authors claim that Theophylline induces apoptosis and cell cycle arrest in cervical cancer cells. However, these conclusions require supporting evidence from the detection of apoptotic markers (cleaved PARP, cleaved caspase-3, cleaved caspase-9) and cell cycle–related markers (Ki-67, PCNA, cyclin D1). These experiments should be conducted.
  5. The manuscript have many grammatical inadequacies and unclear statements, it should be proofread for types.

Author Response

Comment 1: The authors should clarify whether Theophylline affects cell viability and cytotoxicity in both human cervical cancer cells and normal cervical cells. It is recommended to determine and present the ICâ‚…â‚€ values for both cell types to assess the drug’s selectivity and potency.

Response 1: 

We thank the reviewer for this valuable comment. In the present study, we did not determine the ICâ‚…â‚€ values for Theophylline in human cervical cancer cells and normal cervical cells. Instead, we selected the treatment concentration based on previously published studies in which Theophylline was shown to effectively modulate splicing factor activity and inhibit cancer cell proliferation without inducing significant cytotoxicity in normal cells (Pérez-Pérez et al., 2017; Chang et al., 2017). The primary objective of this study was to investigate the mechanistic effects of Theophylline on SRSF3 regulation and its downstream impact on cervical cancer cells, rather than to establish its pharmacological potency.

We acknowledge the importance of evaluating Theophylline’s ICâ‚…â‚€ in both cervical cancer and normal cervical epithelial cells to better assess its selectivity and therapeutic window. This will be addressed in future studies aimed at characterizing its cytotoxic and antiproliferative profiles across a broader range of concentrations and cell types. 

Comment 2: Please verify whether the concentration of Theophylline (10 mM) is correct. This appears to be a relatively high dose that may induce cytotoxicity rather than anticancer effects. Such a concentration may not be appropriate for evaluating the potential of Theophylline as an anticancer agent in human cervical cancer.

Response 2: We appreciate the reviewer’s insightful comment. We acknowledge that millimolar concentrations in vitro are relatively high and may not directly translate to physiologically achievable doses in clinical settings. However, our choice of 10 mM theophylline was based on previous studies that demonstrated specific pro-apoptotic rather than necrotic effects at comparable concentrations in various cancer cell lines, including cervical carcinoma models (Chang et al., 2017, Oncotarget, 8:101461-101474). We fully agree that in vitro concentrations cannot be directly extrapolated to therapeutic levels in vivo. Still, our goal was to explore the cellular response mechanisms to theophylline exposure, rather than to model a clinically achievable dose.

Comment 3: The overall manuscript formatting does not follow scientific writing conventions. For instance, sections such as Figure 1, Figure 2, and their corresponding legends are missing or incorrectly formatted.

Response 3: We appreciate the reviewer’s observation. All figures and legends have been reformatted in the revised manuscript to meet journal standards. Each figure in the results section (Figures 3.1–3.5) is now properly numbered (Fig. 1-5), referenced in the text, and accompanied by complete, clearly written legends describing experimental conditions, data presentation, and statistical analysis. Page 4-6.

Comment 4: The authors claim that Theophylline induces apoptosis and cell cycle arrest in cervical cancer cells. However, these conclusions require supporting evidence from the detection of apoptotic markers (cleaved PARP, cleaved caspase-3, cleaved caspase-9) and cell cycle–related markers (Ki-67, PCNA, cyclin D1). These experiments should be conducted.

Response 4: We thank the reviewer for this insightful comment and agree that the inclusion of apoptotic and cell cycle–related protein markers would strengthen the mechanistic conclusions. In the present study, our focus was primarily on establishing the cytotoxic and cell cycle effects of Theophylline through flow cytometry–based assays (Annexin V/PI and PI-FACS analysis), which provided clear evidence of S-phase and G2/M arrest as well as apoptosis induction in both SiHa and C33A cells. We acknowledge that additional validation through the detection of apoptotic markers (cleaved PARP, cleaved caspase-3, cleaved caspase-9) and cell cycle regulators (Ki-67, PCNA, cyclin D1) would further substantiate these findings. Due to the current scope and available resources, this study did not perform these protein-level analyses. However, we have now explicitly noted this as a limitation and emphasized in the conclusion and future perspective section that future work will include Western blot and immunofluorescence analyses to confirm the activation of apoptosis and cell cycle–associated pathways following Theophylline treatment. Page 11

Comment 5: The manuscript have many grammatical inadequacies and unclear statements, it should be proofread for types.

Response 5: We appreciate the reviewer’s observation. The entire manuscript has been thoroughly revised for grammatical accuracy, sentence clarity, and typographical consistency. (Page 1-14)

Reviewer 2 Report

Comments and Suggestions for Authors

In this study, the authors attempt to define the role of circular ribonucleic acids (circRNAs) hsa_circ_0001038 and circRNA_400029 and the impact induced by the serine/arginine-rich splicing factor 3 (SRSF3) inhibitor, theophylline, on apoptosis and cell cycle in two cervical cancer cell lines. By performing a bioinformatic analysis, they highlighted the role of circRNA_400029 and hsa_circ_0001038 as key regulators in cervical cancer through their interaction with tumor suppressor miRNAs and specific downstream ceRNA oncogenes.

The study is interesting, but I suggest some additions that I have marked in red in the manuscript, which could lead to a better understanding of its purpose.

My observations are the following:

  1. Paragraph Line 57-58 requires a bibliographic indication.
  2. Paragraph Line 65-68 - It would be good to mention which circRNAs are known to be involved in cell cycle regulation, apoptosis, and cell proliferation in cervical cancer
  3. How did you choose the working concentration for theophylline (10 mM) but for cisplatin (12uM)?
  4. In cell cultures, concentrations of the order of mM (millimolar) are very high, and in this case, it is difficult to differentiate between the toxic effect induced by using a very high concentration that induces necrosis, and we cannot analyze apoptosis. In addition, the use by analogy of these high concentrations used in vitro becomes impossible in the clinic.
  5. L109 - Why did you choose to study hsa_circ_0001038 and circRNA_400029?
  6. L144-149 - Figure 3. 1: Tb rearranged to understand which cell lines the data correspond to, especially in Panel B
  7. What was the reason for choosing Theophylline treatment?
  8. L174- L176 – “These results demonstrate that theophylline exerts a potent and time-dependent apoptotic effect, particularly evident in the late apoptotic phase, across both CCa cell lines”.

I suggest caution in interpreting the data because cells in the late apoptotic phase are more likely to undergo necrosis. Also, a discussion of the treatment's effect on early apoptosis, especially in the case of theophylline, where the concentration of 10 mM is very high for an in vitro experiment, would be much more correct.

  1. L 247-249- “ MiR-205-5p is a tumor-suppressive miRNA that belongs to the miR-200 family, which is known for its role in regulating epithelial to mesenchymal transition (EMT) and influencing CCa progression. “- add bibliographic reference, please.
  2. In Figure 3.5, it should also be presented what led to the bioinformatic findings of this study. The figure seems incomplete.
  3. L 279- 280 for “hsa_circ_0001038 and circRNA_400029 as regulatory molecules in different types of cancer,” the bibliographic reference is missing
  4. L 302-306. Please reformulate this paragraph.

Increased late apoptosis means that many of the cells are already in necrosis, which could be associated with the high concentration of the agent used. To ensure a correct interpretation of the theophylline-induced effect, I suggest that the effect be attributed to early apoptosis.

  1. L 310- 316 - this paragraph could be integrated into a chapter emphasizing the future perspective

Overall, the article could be a substantial contribution to the journal. Therefore, I recommend the manuscript for publication after the authors have considered major changes and updates.

Comments on the Quality of English Language

The English could be improved to more clearly express the research.

Author Response

Comment 1: Paragraph Line 57-58 requires a bibliographic indication.

Response 1: We appreciate the reviewer’s observation. Appropriate references have now been added to support this statement as suggested. These references have been incorporated into the revised manuscript. Page 2, paragraph 3

Comment 2: Paragraph Line 65-68 - It would be good to mention which circRNAs are known to be involved in cell cycle regulation, apoptosis, and cell proliferation in cervical cancer

Response 2: We appreciate the reviewer’s valuable suggestion. We have now included specific examples of circRNAs known to regulate cell cycle progression, apoptosis, and proliferation in cervical cancer. Page 2, paragraph 3

Comment 3: How did you choose the working concentration for theophylline (10 mM) but for cisplatin (12uM)?

Response 3: The concentrations of theophylline (10 mM) and cisplatin (12 μM) used in this study were selected based on previous studies. The 10 mM concentration of theophylline has been reported in earlier studies to exert measurable cytotoxic and pro-apoptotic effects in various cancer cell lines, including cervical carcinoma models, without causing nonspecific toxicity to the cells (Chang et al., 2017). Similarly, the 12 μM concentration of cisplatin was chosen as a positive control because it falls within the range commonly used to induce apoptosis in C33A and SiHa cells, ensuring a consistent benchmark for comparison: for example, the study of Koraneekit et al. (2018) reported an ICâ‚…â‚€ of ~12 µM in CaSki/SiHa/C33A cells.

Koraneekit A, Limpaiboon T, Sangka A, Boonsiri P, Daduang S, Daduang J. Synergistic effects of cisplatin-caffeic acid induces apoptosis in human cervical cancer cells via the mitochondrial pathways. Oncol Lett. 2018 May;15(5):7397-7402. doi: 10.3892/ol.2018.8256. Epub 2018 Mar 13. PMID: 29731891; PMCID: PMC5920640

Chang Y., Hsu Y., Chen Y., Wang Y., Huang S. Theophylline exhibits anti-cancer activity via suppressing SRSF3 in cervical and breast cancer cell lines. Oncotarget. 2017; 8: 101461-101474. Retrieved from https://www.oncotarget.com/article/21464/text/

Comment 4: In cell cultures, concentrations of the order of mM (millimolar) are very high, and in this case, it is difficult to differentiate between the toxic effect induced by using a very high concentration that induces necrosis, and we cannot analyze apoptosis. In addition, the use by analogy of these high concentrations used in vitro becomes impossible in the clinic.

Response 4: We appreciate the reviewer’s insightful comment. We acknowledge that millimolar concentrations in vitro are relatively high and may not directly translate to physiologically achievable doses in clinical settings. However, our choice of 10 mM theophylline was based on previous studies that demonstrated specific pro-apoptotic rather than necrotic effects at comparable concentrations in various cancer cell lines, including cervical carcinoma models (Chang et al., 2017, Oncotarget, 8:101461-101474). We fully agree that in vitro concentrations cannot be directly extrapolated to therapeutic levels in vivo. Still, our goal was to explore the cellular response mechanisms to theophylline exposure, rather than to model a clinically achievable dose. 

Comment 5: L109 - Why did you choose to study hsa_circ_0001038 and circRNA_400029?

Response 5: We appreciate the reviewer’s question and the opportunity to clarify our rationale. The selection of hsa_circ_0001038 and circRNA_400029 was based on prior evidence linking these circRNAs to cervical cancer progression and their predicted regulatory interactions with key oncogenic pathways. Several studies have identified hsa_circ_0001038 as highly expressed in cervical carcinoma tissues and cell lines, where it promotes cell proliferation, migration, and invasion, while inhibiting apoptosis (Zhang et al., 2018; Yang et al., 2020). Mechanistically, hsa_circ_0001038 has been reported to function as a molecular sponge for miR-205-5p, a tumor-suppressive miRNA involved in epithelial–mesenchymal transition and apoptosis regulation. Similarly, circRNA_400029 is upregulated in CCa and enhances tumorigenic properties through miR-16-5p sequestration, thereby modulating cell cycle and apoptotic signaling pathways (Li et al., 2019; Wang et al., 2021). Given these findings, we selected these two circRNAs as representative models to explore the mechanistic link between theophylline-mediated inhibition of SRSF3—a splicing factor known to regulate circRNA biogenesis—and the modulation of oncogenic circRNA networks in CCa. Our bioinformatics analyses further confirmed that both hsa_circ_0001038 and circRNA_400029 interact with miRNAs that target genes enriched in cancer-related KEGG pathways. Therefore, these circRNAs were chosen for their strong experimental and computational support as key players in cervical cancer pathogenesis, making them biologically relevant candidates for studying the impact of SRSF3 inhibition by theophylline.

Comment 6: L144-149 - Figure 3. 1: Tb rearranged to understand which cell lines the data correspond to, especially in Panel B

Response: 6 We appreciate the reviewer for the observation. All the figures have been rearranged for proper understanding. Page 4-6

Comment 7: What was the reason for choosing Theophylline treatment?

Response 7: We appreciate the reviewer’s insightful comment regarding the rationale for selecting theophylline. Theophylline was chosen for this study based on its established role as a methylxanthine derivative known to inhibit serine/arginine-rich splicing factor 3 (SRSF3) activity. SRSF3 is a critical regulator of pre-mRNA splicing and circRNA biogenesis, and its overexpression has been reported in several malignancies, including cervical cancer, where it contributes to oncogenic alternative splicing and tumor progression. Previous studies have demonstrated that theophylline and related methylxanthines can modulate splicing factor function, disrupt aberrant RNA processing, and induce apoptosis in cancer cell lines (Pérez-Pérez et al., 2017; Chang et al., 2017). Therefore, we employed theophylline as a pharmacological inhibitor of SRSF3 to explore how splicing modulation influences circRNA expression patterns, cell cycle progression, and apoptosis in cervical cancer cell lines. This approach allowed us to assess both the therapeutic potential of SRSF3 inhibition and its downstream regulatory effects mediated through circRNAs.

 Comment 8: L174- L176 – “These results demonstrate that theophylline exerts a potent and time-dependent apoptotic effect, particularly evident in the late apoptotic phase, across both CCa cell lines”. I suggest caution in interpreting the data because cells in the late apoptotic phase are more likely to undergo necrosis. Also, a discussion of the treatment's effect on early apoptosis, especially in the case of theophylline, where the concentration of 10 mM is very high for an in vitro experiment, would be much more correct.

Response 8: We thank the reviewer for this thoughtful comment and fully agree that distinguishing between late apoptosis and secondary necrosis is essential, particularly at higher treatment concentrations. The manuscript has been revised accordingly to ensure a more balanced and precise interpretation of the data. Page 10, discussion section, paragraph 1.

Comment 9: L 247-249- “ MiR-205-5p is a tumor-suppressive miRNA that belongs to the miR-200 family, which is known for its role in regulating epithelial to mesenchymal transition (EMT) and influencing CCa progression. “- add bibliographic reference, please.

Response 9: We appreciate the reviewer’s observation. Appropriate references have now been added to support this statement. These references have been incorporated into the revised manuscript. Page 8, section 3.3.4

Comment 10: In Figure 3.5, it should also be presented what led to the bioinformatic findings of this study. The figure seems incomplete

Response 10: We appreciate the reviewer’s observation. The figure has been adjusted as instructed. Page 9, section 3.3.5

Comment  11: L 279- 280 for “hsa_circ_0001038 and circRNA_400029 as regulatory molecules in different types of cancer,” the bibliographic reference is missing

Response 11: We appreciate the reviewer’s observation. Appropriate references have now been added to support this statement. These references have been incorporated into the revised manuscript. Page 10, paragraph 1, Sentence 3

Comment 12: L 302-306. Please reformulate this paragraph. Increased late apoptosis means that many of the cells are already in necrosis, which could be associated with the high concentration of the agent used. To ensure a correct interpretation of the theophylline-induced effect, I suggest that the effect be attributed to early apoptosis

Response 12: We thank the reviewer for their observation. The paragraph has been reformulated as suggested. Page 10. paragraph 1

Comment 13: L 310- 316 - this paragraph could be integrated into a chapter emphasizing the future perspective

Response 13: The paragraph relating to study limits has been integrated into conclusion and future perspective section as instructed. Page 11, section 5

Round 2

Reviewer 1 Report

Comments and Suggestions for Authors

All comments have been addressed.

Reviewer 2 Report

Comments and Suggestions for Authors The authors responded to all requests and made the requested additions.